# Examining the quality of care across the continuum of maternal care (antenatal, perinatal and postnatal care) under the expanded free maternity policy (Linda Mama Policy) in Kenya: a mixed-methods study

Boniface Oyugi [1,2] Zilper Audi-Poquillon [3] Sally Kendall [2]
Stephen Peckham [2]

¹Western Heights, The Mint Nairobi, M and E Advisory Group, Nairobi, Kenya
²Centre for Health Services Studies, University of Kent, Canterbury, UK
³Department of Health Policy, The London School of Economics and Political Science, London, UK

**Correspondence to**
Zilper Audi-Poquillon;
z.a.audi-poquillon@lse.ac.uk

## ABSTRACT

**Background** Kenya still faces the challenge of mothers and neonates dying from preventable pregnancy-related complications. The free maternity policy (FMP), implemented in 2013 and expanded in 2017 (Linda Mama Policy (LMP)), sought to address this challenge. This study examines the quality of care (QoC) across the continuum of maternal care under the LMP in Kenya.

**Methods** We conducted a convergent parallel mixed-methods study across multiple levels of the Kenyan health system, involving key informant interviews with national stakeholders (n=15), in-depth interviews with county officials and healthcare workers (HCWs) (n=21), exit interview survey with mothers (n=553) who utilised the LMP delivery services, and focus group discussions (n=9) with mothers who returned for postnatal visits (at 6, 10 and 14 weeks). Quantitative data were analysed descriptively, while qualitative data were analysed thematically. All the data were triangulated at the analysis and discussion stage using a framework approach guided by the QoC for maternal and newborns.

**Results** The results showed that the expanded FMP enhanced maternal care access: geographical, financial and service utilisation. However, the facilities and HCWs bore the brunt of the increased workload and burnout. There was a longer waiting time for the initial visit by the pregnant women because of the enhanced antenatal care package of the LMP. The availability and standards of equipment, supplies and infrastructure still posed challenges. Nurses were multitasking and motivated despite the human resources challenge. Mothers were happy to have received care information; however, there were challenges regarding respect and dignity they received (inadequate food, over-crowding, bed-sharing and lack of privacy), and they experienced physical, verbal and emotional abuse and a lack of attention/care.

**Conclusions** Addressing the negative aspects of QoC while strengthening the positives is necessary to achieve the Universal Health Coverage goals through better quality service for every woman.

## STRENGTH AND LIMITATIONS OF THIS STUDY

⇒ This is the first study to explore the optimal quality of care (QoC) across the continuum of maternal care (antenatal, perinatal and postnatal care) under the expanded free maternity policy (FMP) in Kenya using the QoC for Maternal and Newborn—a monitoring framework for network countries.

⇒ The use of a mixed-methods approach in this study permitted complementarity, convergence and triangulation of the qualitative and quantitative data to deepen the policy description and analysis, hence attenuating the weaknesses of the singular methods.

⇒ While the results may not be generalisable beyond the study county (area) because of the heterogeneity of the counties, this study identifies significant contextual factors that may have influenced the patterns of implementation and the findings, which are transferable (enhanced transferability) to other 47 counties in the counties and can be used to interpret the implications of the results in other settings.

⇒ There could be many other unidentified QoC elements from this study, particularly other county-specific issues, but the findings could be considered the first step in exploring and compiling the existing knowledge about the global situation.

⇒ This study could be particularly informative for policymakers as a guide to effective evidence-based interventions that can be adopted to strengthen the implementation of the FMP in the country.

## INTRODUCTION

There are nearly 287 000 maternal deaths due to preventable pregnancy and childbirth-related complications happening globally (translating to almost 800 maternal deaths every day or one every 2 min).[1] Low-income and middle-income countries and low-income countries, especially those in

sub-Saharan Africa, such as Kenya, are the most affected because of barriers to accessing maternal services (such as low quality of care (QoC), poor socioeconomic conditions, poor infrastructure and lack of well-trained healthcare professionals).[2–4] While Kenya's maternal and child health status has significantly improved in the last decade, the current maternal mortality ratio of 530 deaths per 10 000 live births is significantly higher than the world average of 223 maternal deaths per 100 000 live births,[1] as is the neonatal mortality rate of 21 deaths per 1000 live births which is higher than the world average of 18 deaths per 1000.[5 6] Approximately 7300 women still die every year, making up 15% of all deaths among women of reproductive age, with both mothers and neonates dying from preventable pregnancy-related complications.[7] One in 76 women in Kenya is at risk of dying from pregnancy complications.[8]

Reducing and eliminating pregnancy-related mortality, ending preventable newborn and child mortality and achieving Universal Health Coverage (UHC) remain crucial targets and priorities for realising the Sustainable Development Goals (SDGs) in Kenya. Various reforms in the health sector in Kenya have sought to achieve the above SDG targets by reducing catastrophic expenditure on maternity care and improving the quality of healthcare service delivery.[9–13] One such reform, the user fee waiver for maternity and primary healthcare (PHC) services, was initiated by the government of Kenya, in June 2013.[9] This reform, while significant, faced challenges of poor service delivery due to inadequate preparation before the implementation and a lack of adequate systems to verify the quality of care provided and the reimbursement claims from the hospitals to the government.[14]

Subsequently, to overcome these challenges, the country transitioned to a new expanded free maternity policy (FMP) in 2017 to provide access to maternal services to all pregnant women in an expanded network of providers, including private, faith-based and all level 3–6 public institutions.[15] The expanded FMP was called Linda Mama (LM) (Swahili for 'caring for the mother'), and was managed through the National Hospital Insurance Fund (NHIF) to overcome challenges from the previous policy by enhancing administrative efficiency, ameliorating the reimbursement logistical challenges, creating a longer-term financing sustainability and easing legal hurdles.[16] Besides, it aimed to improve access to quality maternal and child services and reduce inequalities, thereby advancing the country's agenda of UHC.[15 17] The benefits package of the expanded policy captured both inpatient and outpatient services (including more antenatal services, delivery, postnatal care (PNC) and referrals of emergencies of pregnancy-related conditions and complications) for the mother and the newborn up to a year.[18 19]

Being part of the reform linked to the UHC agenda, there were three facets targeted for improvements: population, services and direct costs,[20] envisaging that every person would have access to the entire range of quality health services and care they needed, whenever and wherever they needed them, without financial hardship.[21 22] The LM policy was mainly implemented to achieve the three facets. However, following the implementation of the two free maternity policies, researchers have focused on mostly understanding two facets (population and cost), through studies focused on the policy's immediate and trend effect,[23] its impacts on mortality and utilisation of services,[24–27] out of pocket expenditure,[28] policy formulation and implementation elements[15–17] and the cost-benefit analysis.[27] While there has been an attempt to look at the services, the quality of services and care aspects from both policies have not been conclusive and a gap remains. For instance, one study evaluated satisfaction with the delivery services under FMP.[29] It showed that the mothers who benefited from the services were satisfied with different components such as communication by the healthcare workers (HCWs), staff availability in the wards and delivery rooms, and supplies availability, but were also unsatisfied with cleanliness, consultation time and privacy in the wards. Another study evaluating the utilisation of the free maternity services implemented in 2013 among women living in Kibera slums in Nairobi showed that mothers positively perceived the distance to the facility and shorter waiting time, in addition to patients facing bad providers' attitudes.[30] Yet, another study that evaluated disrespectful maternal care under the policy in Kisii and Kilifi counties showed that mothers experienced disrespectful maternal care throughout the maternity process, and it appeared even more significant among women who were poor, young or had children with disabilities.[31] All three studies on quality have focused on one aspect of quality: the outcome (from the patient perspective), leaving out other quality dimensions that researchers[32 33] have discussed: structure, process and outcome.

Therefore, the quality-of-service facet is yet to be fully explored. One study evaluated the characteristics associated with the QoC of the initial assessment for pregnant mothers, intrapartum and postpartum and newborn care (continuum of care) not under the FMP but in the country context using service provision data and the finding was that a sustained focus on the QoC along the maternity care continuum was imperative for the mothers and their newborns and policymakers (in distributing resources to improve the areas of the continuum).[34] Increasing service coverage alone is unlikely to produce better health outcomes without paying attention to the QoC provided. The LM policy seeks to be a high-quality health intervention that optimises maternal care in the Kenyan context by consistently delivering and giving care that enhances or maintains maternal and neonatal outcomes, and that is valued and trusted by everyone since it responds to a changing population's needs.[35] Maternal care under LM policy envisages enhancing the degree to which maternal services received by clients increase the likelihood of desired health outcomes consistent with current professional knowledge and are effective, safe, people-centred,

timely, equitable, integrated and efficient.[36] Therefore, exploring the optimal quality of maternal care and outcomes from the LM policy would be imperative. This study examines the QoC across the continuum of maternal care (antenatal, perinatal and postnatal care (PNC)) under the LM policy in Kenya.

## METHODS
### Study design
We utilised the convergent mixed-methods design, specifically the parallel-database variant in this study,[37] using qualitative and quantitative data that were collected and analysed in tandem and then compared and combined to better understand the QoC across the continuum of maternal care (antenatal, perinatal and PNC) under the expanded FMP in Kenya.

### Framework for analysis
Recognising that quality cannot be measured by itself,[38] in this study, we conceptualised quality from the Donabedian perspective, broadly classifying quality as structure, process and outcome dimensions,[32 33] which can be identified, measured and attributed to healthcare. Akachi and Kruk[39] provide more details on measuring changes in the QoC, emphasising the inclusion of user experience as a measure of outcomes in the quality assessment. With these two refined aspects, we broadly define structure indicators as pointers which are inputs to or characteristics of health; process indicators as gauges of appropriate or inappropriate care in a targeted population, which are 'consistent with current professional knowledge'; and outcome indicators as the measures of both improved or deteriorated health and attributed to medical care[38 39]

(see figure 1). Our data collection methods and tools were designed to comprehensively examine all aspects of QoC across the continuum of maternal care (antenatal, perinatal and PNC) under the free maternity policy in Kenya. Broadly, our analysis converges all the concepts using the QoC for Maternal and Newborn—a monitoring framework for network countries,[40] which draws concepts from the earlier framework as proposed by the WHO.[41]

### Study setting
The study was conducted across multiple levels within the Kenyan health system. The Kenyan health system is pluralistic in the provision and financing of services and is organised into six levels of care. Level 1 forms the community units overseen by community health workers whose role is providing promotive services (health education, treating minor ailments and identifying cases that require referral to health facilities),[42] and both level 2 (dispensaries) and level 3 (health centres) provide PHC services in addition to coordinating the community in their areas of jurisdiction. Levels 4 and 5 offer curative services as county secondary referral facilities, with some being training centres, while level 6 are semiautonomous tertiary facilities offering specialised care and serving as training institutions.

At the national level, we included the Ministry of Health (MoH), the NHIF and development partner agencies involved in the expanded FMP. At the county level, this study was conducted in Kiambu County in Kenya. While this study is part of a larger study, we purposefully chose Kiambu County because of its sociodemographic characteristics, health indicators and population size.[43–45] It is the second-most populous county in Kenya after Nairobi

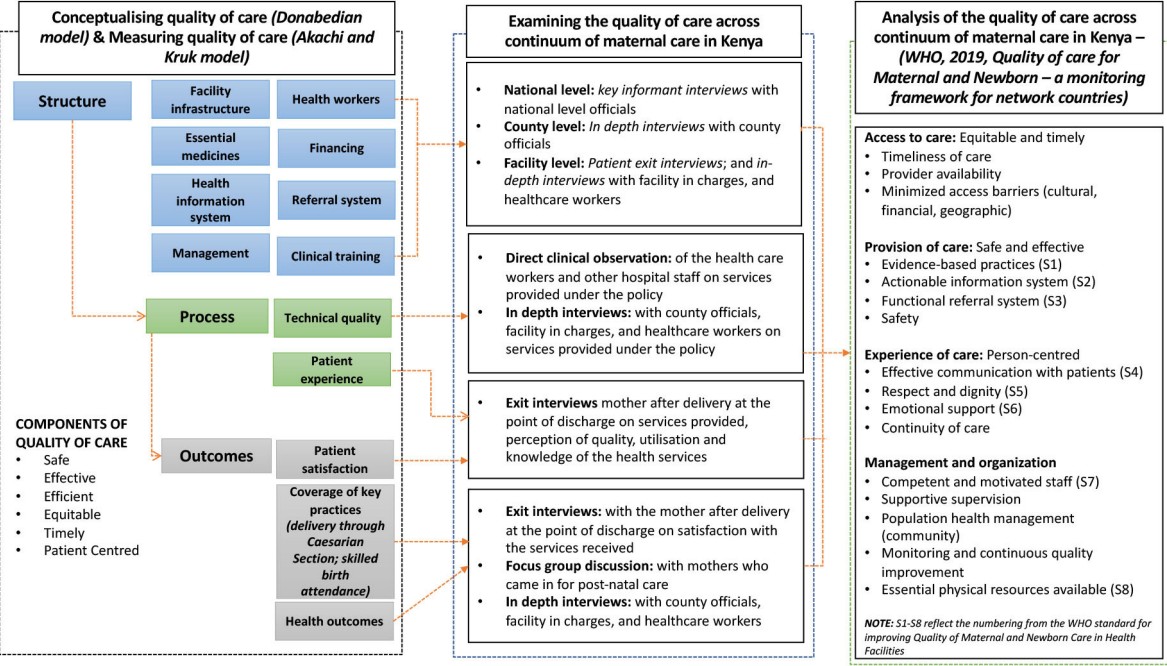

**Figure 1** Combined frameworks used in this study for examining the quality of care across the continuum of maternal care (antenatal, perinatal and postnatal care) under the free maternity policy in Kenya.

City County, with a population of 2 417 735: 49.1% male and 50.59% female,[43] 26.9% of the population in Kiambu are female of reproductive age (15–49 years),[44] 89.2% of births in the county happen in a health facility, 98.2% of births provided by a skilled provider, 67% of women aged 15–49 who had a live birth had 4+ antenatal visits and 89% of women aged 15–49 had a postnatal check during the first 2 days after birth.[5] While these statistics are slightly higher than the national average, they have not translated to quality care. Research has shown that primary care facilities with a low delivery volume have very low-quality delivery care, indicating crucial deficiencies in infrastructure and staffing, routine and emergency care practices, and referral systems.[46] A majority of the facilities providing care in Kiambu are primary care facilities (70 tier 2—dispensaries and tier 3—health centres) with low volumes compared with the secondary facilities (13 tier 4—hospitals and 1 tier 5—intercounty facility) with a high volume.[47] With the secondary care facilities receiving a higher population for care from the neighbouring counties and the locals,[48] they are bound to be stretched beyond the limit, hence the potential for challenges in the QoC. Kiambu faces challenges with the interventions to address maternal health, such as referral systems that work, family planning, access to safe abortion services, availability of skilled health workers and accessible health facilities[49–51] despite being cosmopolitan. With the majority of the population in Kiambu being urban,[43] which is facing overcrowding, there is a potential for the urban averages of maternal mortality to become either closer or worse than rural averages. Furthermore, Kiambu County was purposively selected for this study because the county has been shown to pose higher trends in maternal mortality compared with other counties around Nairobi from the Central Region.[52]

We purposefully selected three study facilities: a level 3 facility (considered a low volume—few numbers of clients), a level 4 facility (medium volume) and a level 5 facility (high volume). We chose these facilities in consultation with the county team to provide nuanced, unique subcounties dynamics given their richness in information and characteristics (see table 1).

### Study population, sampling and data collection
The study population used in this study were in four categories, as summarised in table 1. We collected data between November 2018 and September 2019. This included exit interviews (EIs), focus group discussions (FGDs), in-depth interviews (IDIs) and key informant interviews (KIIs).

The first group of respondents, staff from the MoH, NHIF and development partners, were purposely selected based on their level of involvement in the expanded FMP. These respondents participated in KIIs with one researcher (BO), which were done in English, using KII guides developed to capture the experience of the formulation and implementation of the expanded FMP. All the KIIs (n=15) were conducted in Nairobi, and were audiotaped following participants' consent using audio recorders. Each KII lasted between 45 and 60 min.

The second category was comprised of purposively selected respondents with knowledge of and experience in the implementation of the expanded FMP at the county (meso) level. This group included county and subcounty level officials from the County Department of Health; and the facility (micro) level (including facility in-charge, HCWs in charge of/offering maternal care/services and other cadres of hospital workers) (table 1). These respondents participated in IDI with one researcher (BO). The IDIs (n=21) were conducted in English using two semi-structured guides (each for the county and health facility participants) developed to capture the experience of implementing the expanded FMP. The construct validity of the two semi-structured guides was tested in the non-participating facility to check for ambiguity and flow of the questions. All the IDIs (save for one conducted at the place of convenience for the participant) were conducted at the participants' places of work and were audiotaped using audio recorders after obtaining their consent. Each IDI lasted between 30 and 60 min. The KIIs and IDIs were stopped when no new information, further dimensions, nuances or insights were forthcoming (ie, when we attained meaning saturation).[53] At this point, we were confident that we had fully understood the issue under discussion.

The third group comprised of EIs with mothers who had delivered in the three hospitals and were discharged home. The sample size of the mothers was estimated at 553 using the formula proposed by Gorstein et al.[54] A detailed discussion of the sample criteria and dynamics across the three selected facilities has been published elsewhere.[55] Four trained data collectors, supervised by one researcher (BO), conducted the EIs with the women. The design of the EI utilised a structured questionnaire, adapted from Dalinjong et al,[56] to elucidate the socio-demographic information of the women, health and related services received at the facility (perception of the quality of maternal care that the mothers received during delivery and antenatal care (ANC) care, experiences with the FM policy). The conduct of the EIs ensured that one researcher (BO) introduced the data collectors to the administration and the maternity department heads of the three facilities; then, each morning of the interview, they identified the mothers who had been discharged (using bed numbers) and were waiting to return home. With the number of mothers identified per day, we generated a random sample using Stat Trek's Random number generator,[57] which was used to identify mothers for the EI. The mothers were then invited to participate in the study, and interviews were conducted until we reached the intended sample size. We took each mother through the information sheet, and only when they were comfortable participating did we give them the consent forms. One mother declined to participate (and we eliminated two entries at the analysis stage for lacking complete information).

**Table 1** Hospital characteristics and study population

| | Level 3 hospital (hospital A) | Level 4 hospital (hospital B) | Level 5 hospital (hospital C) |
|---|---|---|---|
| Hospital characteristics | | | |
| Bed and cots capacity* | 10 | 46 | 289 |
| Number of staff† | 35 | 115 | 262 |
| Estimated annual deliveries‡ | 1076 | 5635 | 9152 |
| Estimated annual outpatient care‡ | 88 829 | 156 108 | 281 379 |
| Estimated annual inpatient care‡ | 764 | 7223 | 14 205 |
| Hospital participants in the study | | | |
| EIs | 42 | 170 | 338 |
| FGDs | 3 | 3 | 3 |
| IDIs | 7 | 5 | 6 |
| Facility-level managers | 1 | 3 | 2 |
| Department in-charges | 1 | 1 | 1 |
| Nursing officers | 4 | 0 | 1 |
| Accounting/clerical officers | 1 | 1 | 2 |
| County participants (IDI) | | | 3 |
| Senior-level managers | | | 1 |
| Middle-level manager | | | 2 |
| National participants (KIIs) | | | 15 |
| Ministry of Health officials | | | 5 |
| NHIF officials | | | 3 |
| Development partners | | | 7 |

Estimates for annual delivery, outpatient care and inpatient care were for the financial year July 2018–June 2019. The outpatient total is an aggregate of both new and revisits.
*Kenya Master Health Facility List.[101]
†In-depth interview with health facility in-charges of the individual facilities.
‡Kenya Health Information System (KHIS) for aggregate reporting.[102]
EIs, exit interviews; FGDs, focus group discussions; IDIs, in-depth interviews; KIIs, key informant interviews; NHIF, National Hospital Insurance Fund.

The final category included FGD with nine groups of mothers (ranging from 5 to 12 mothers) purposively selected based on a common interest: mothers who had had a skilled delivery in a hospital setting and had come to the study sites for the 6, 10 or 14 week postnatal visits. One researcher (BO) conducted all nine FGDs in Swahili (given the different levels of knowledge of the participants) using an FGD guide developed in reference to the gaps that had arisen from the EIs. The mothers in the FGD were recruited from the child welfare clinic of the three facilities when they brought their children for routine vaccination. The FGDs in each facility were organised with the help of a nurse from the maternity department. We engaged the mothers as the children received their vaccinations and asked if they would participate in the study. All the FGDs were conducted in a prebooked room at the facilities and were audiotaped following participants' consent using audio recorders. Each FGD lasted between 45 and 90 min.

## Data management and analysis

Quantitative data from the EI were manually entered from the structured questionnaire into the Excel software by one researcher (BO), who cleaned, checked the data for completeness, and then exported it to STATA V.15 for coding and analysis. The socio-demographic characteristics and the elements of quality were analysed descriptively using proportions.

All recorded FGDs were translated from Swahili to English, while the IDIs were transcribed verbatim in English. All transcripts were compared against their respective audio files by BO for transcription and translation accuracy. All the validated transcripts were imported into NVivo V.12 for coding, guided by the topic areas of quality of maternal healthcare. We used a framework approach to analyse the data guided by the QoC for Maternal and Newborn—a monitoring framework for network countries.[40] This approach included systematic sifting, sorting, coding and charting data into key issues

and themes.[58] One researcher (BO) familiarised himself with the data through immersion and repeatedly read and reread the transcripts. He then developed codes deductively from the conceptual framework and applied the codes to interpret segments in the transcripts that were important. The study team members (SK and SP) reviewed and discussed the initial coding framework, and any discrepancies were appropriately reconciled. The final coding framework was applied by (BO) to the data, and later, the data was charted to allow the emergence of themes through comparisons and interpretations.

To enhance the interpretive rigour, we ensured credibility (also referred to as internal validity) through the convergence of evidence of the two methods utilised and triangulation (investigator, theoretical and methodological) of data at the interpretive stage.[59]

## Ethics consideration

We received written permission from the county government and all the hospitals to conduct the study. Before starting the interviews, we obtained written and oral informed consent from the potential participants. All the study participants were presented with information sheets on the conduct of the study, the researchers involved, the purpose of the study, the right to withdraw and measures of confidentiality ensured before they gave their written informed consent. Participants were informed that data would be reported in an aggregated format, and anonymity would be ensured in storing and publishing the study's findings.

## Patient and public involvement

Patients and/or the public were not involved in the design, or conduct, or reporting or dissemination plans of this research. We intend to disseminate these research findings to the public through a summarised press article or brief, social media and the websites of authors' institutions.

## RESULTS

The results on the quality of maternal care in this study were presented using the WHO-proposed monitoring logic model from the perspective of the policy implementers and users. Results are presented in four broad domains: access to care (equitable and timely), provision of care (safe and effective), management and organisation and care experience. A summary of the results is presented in table 2.

## Element 1: access to maternal care services under the expanded FMP (equitable and timely)
### Minimised access barriers (cultural, financial and geographic)
*The expanded FMP enhanced maternal care access elements (geographical, financial or utilisation of services).* For instance, due to the policy, there was an increase in the utilisation of maternal services (delivery and ANC). Findings from EIs showed that most mothers across the study sites (99.09%,

n=545) visited a hospital for maternal health services during their pregnancy (online supplemental appendix 1). Furthermore, IDI showed that more mothers (than previously) were confident in seeking skilled services rather than remaining at home.

> … mothers who could not come, now they are coming [to the facility]. And there is also a change in the number of deliveries we used to have before and now. (R009, Nursing Officer)

The respondents highlighted the importance of the policy's impact on equity. They noted that the enhanced identification strategies for the mothers implemented under the expanded FMP, saw increased access to services among vulnerable populations such as street children, orphans and adolescents. Moreover, they observed that the policy enhanced equity (as women in the rural and urban areas received uniform services), and the women had better financial access to the services (as these services were available for free).

*However, the facilities and HCWs were bearing the burden of increased numbers of mothers seeking LM care.* As noted by most respondents, facilities were bearing the brunt of the increased number of mothers due to LM, which resulted in both space shortages and increased workload, impacting the quality of care. The workload was further exacerbated by the nature of work in the public facilities where the HCWs had no choice but to serve the mothers and meet the required utilisation targets. Because, the facilities were working way beyond their abilities to manage the workload, it also resulted in HCWs experiencing some burnout:

> We work extra hours … you will find each care provider is serving more than they should, so the issue of burnout is also coming up. (R019, Facility-Level Manager)

*There was an altered perception among women, leading to a preference for higher-level facilities.* There was an increased workload in higher-level facilities caused by the mother's perception that these facilities have specialist healthcare professionals, unlike the lower-level dispensaries or community centres. The women believed that higher-level facilities were better equipped to deal with complications than the lower-level hospitals:

> … sometimes you ask them, 'Why have you decided to come here?' 'Because here, people who will attend to me are qualified'. …But they say outside there, anybody can attend to you. (R014, Nursing Officer).

## Additional maternal determinants of care and the timeliness of care
*There was a positive perception about the time taken to seek care and the waiting time.* A majority of the women visited a public facility (92%); and had a positive perception about the time taken to the facility and the distance to the hospital. Women who visited hospitals A (45.24%), B

**Table 2** Summary of the quality of maternal care results

| Domain | Subdomain | Positive result | Negative result |
|---|---|---|---|
| Element 1: access to maternal care services under the expanded FMP (equitable and timely) | Minimised access barriers (cultural, financial, geographic) | The expanded FMP enhanced maternal care access elements (geographical, financial or utilisation of services). | However, the facilities and HCWs were bearing the brunt of the burden of increased numbers of mothers seeking LM care (workload and burnout). |
| | | | There was an altered perception among women, leading to a preference for higher-level facilities. |
| | Additional maternal determinants of care and the timeliness of care | The distance to the hospital was perceived as normal (okay for the patients) and the preferred choice of transport to the facility was public transport. | There was a longer waiting time for the initial visit by the pregnant women due to the enhanced ANC package of the expanded FMP. |
| | | All the three hospitals had a proper waiting area. | |
| | | There was a positive perception about the time to seek care and the waiting time. | |
| | Provider availability | | There were problems of struggling to employ specialists and other HCWs staffing challenges. |
| Element 2: provision of care (safe and effective) | Functional referral system | | Fewer women were being referred, but they had a better perception of services received during referral. |
| | | | The facilities' lack of equipment, theatre, NBU and blood were the main reasons for referrals. |
| | Safety | Because of the policy, the facilities were managing complications better. | HCWs were reducing the time they allocate per mother. |
| Element 3: management and organisation | Availability of essential physical resources | The policy has improved the availability and standards of equipment and supplies. | Despite progress, some infrastructure, commodities and supplies are still a challenge to some facilities. |
| | | The facilities had improved infrastructure due to LM. | |
| | | Enhanced facility resources and facility characteristics. | |
| | Competent and motivated staff | Mothers have a strong positive perception of healthcare delivery characteristics by the HCWs. | There were some causes of demotivation and dissatisfaction among HCWs. |
| | | Nurses are multitasking and handling many roles among the challenge of human resources. | |
| | | HCWs are adequately motivated to work despite the challenges. | |
| | | HCWs' source of motivation was more than just money. | |
| | Monitoring and continuous quality improvement | Nurses monitor the quality of care provided through partographing and charting labour progress, though they face challenges. | |

Continued

**Table 2** Continued

| Domain | Subdomain | Positive result | Negative result |
|---|---|---|---|
| Element 4: experience of care | Effective communication with the patients | Mothers perceived and experienced the positive interpersonal qualities of the HCWs. | Inadequate preparation for birth by the HCWs. |
| | | Mothers were happy to have received information about emergency/procedures and training on breastfeeding, family planning and baby care. | The lack of proper education and communication on expectations. |
| | Respect and dignity | | Food was perceived as inadequate in some hospitals. |
| | | | Overcrowding and bed-sharing led to a lack of privacy (congestion) and a lack of essential equipment and supplies, altering the QoC. |
| | Emotional support | | Women were experiencing physical, verbal and emotional abuse. |
| | | | Some mothers experienced a lack of attention/care, negligence, and unhygienic practices from the HCWs and support staff. |

ANC, antenatal care; FMP, free maternity policy; HCWs, healthcare workers; LM, Linda Mama; NBU, newborn unit; QoC, quality of care.

(51.18%), C (46.75%) and overall (48.00%)) noted that they took 30 min to 1 hour to seek delivery services and they perceived the time to be short (online supplemental appendix 1).

A majority of the respondents (61.64%) perceived the distance to the hospital was normal and suitable for the patients. The preferred choice of transport to the facility was public transport (40.73%) (online supplemental appendix 1). Additionally, all three hospitals had a proper waiting area. Most of the women were content with the opening hours of the facilities, and they considered the waiting time before being attended to as short (43.09%) (online supplemental appendix 1).

*There was a longer waiting time for the initial visit by the pregnant women due to the enhanced ANC package of the expanded FMP.* The initial ANC profile included blood tests (for haemoglobin levels, blood group, rhesus and serology), screening for tuberculosis, HIV testing and counselling, urinalysis, preventive services (such as deworming, intermittent preventive treatment for malaria, iron and folate supplementation) and prevention of mother to child transmission. All these were conducted at the same laboratory as other patients in the hospitals; resulting in longer wait for results:

… for the first visit (they) will report here at 8:00 (am) and … get out of this place as late as 3:00 (pm) … because when they come … if it's lab everybody is there, the people who are coming for outpatient services are queuing there (too) … the rebate for the first visit (ANC) … covers up a lot. (R002, Clerical Officer)

### Provider availability
*There were problems of struggling to employ specialists and other HCWs staffing challenges.* The facility in-charges noted that they had a challenge of hiring specialist nurses to take care of the growing numbers, which had been exacerbated by the lack of specialised units in some facilities:

… we could not set up a neonatal ward for lack of a neonatal nurse … (yet) we get so many babies, and with that influx, we could still get some babies … (R020, Facility-Level Manager)

One facility in-charge noted that while the facilities had installed an ultrasound machine to meet the needs of the pregnant mothers, there was a gap in trying to identify the person to operate it and sustainably pay the staff.

The staffing challenge, particularly in the lower-level facilities, was hard to deal with because of the rules of staffing, where, despite the high number of mothers, the number of staff cannot exceed a certain limit:

… I think it's not because of Linda Mama, I think it's because of how it has been, we have been a level 3, although they said they would add us people. But you see they cannot exceed the number of staff in a level 3. If it were a level 4, they would increase. (R007, Department In-Charge)

### Element 2: provision of care (safe and effective)
#### Functional referral system
*Fewer women are being referred, but they have a better perception of services received during referral.* While referral of emergency

cases is essential in preventing complications, results from EI showed that only 10.73% (n=59) of all the women interviewed, had been referred for additional care. Most had been referred from level 3 facilities (n=26), using an ambulance (n=22) or public means (n=15). They were mainly accompanied by their husbands (n=27), relatives (n=23) or health workers (n=21) either as an individual or both at the same time (online supplemental appendix 2). A majority of the mothers' companions had knowledge of emergency management (n=47), were allowed to stay in the hospitals (n=33) and were warmly received at the hospitals (n=19) during the referral (online supplemental appendix 2).

The women in the FGDs perceived that the maternal services provided by the mothers had improved because of the LM policy, leading to a reduction in referrals:

R3: I can say the services are good because nowadays we don't run to (referral hospital) the way we used to. So, this hospital has been good, it has been helpful to us. (Woman in FGD003).

*The lack of equipment was the main reason for referral, and most women sought their own referral means from the hospital.* From the EI, the referred mothers noted that lack of equipment, theatre, newborn unit (NBU) and blood (n=16) were the leading cause of referral, followed by foetal distress (n=7) (online supplemental appendix 3). While HCWs indicated that the county and facilities provide some form of referral transport for mothers, the referred mothers reported seeking their own transport means for referral. These mothers perceived this situation to be both expensive and dangerous for their health and safety, especially in unplanned emergencies.

R5: … they (health workers) told me there's no vehicle, and they insist, 'Look for a vehicle quickly so she can be referred' …now to do it fast and you don't have money … I really suffered; R8: … if a mother delivers now, (and)…is going to (a referral facility) and you know the road there is not good and someone has been stitched up down there (episiotomy) … when going there the stitches might be undone … (Women in FGD009)

### Safety
*Because of the LM policy, the facilities were managing complications better.* HCWs and hospital administrators acknowledged that the policy improved the facilities' management of complications. The policy objectives incentivised them:

… for example, she (patient) comes up with a chronic infection, which means the administration will spend more money buying an expensive drug for her. But you see, the moment she comes on time, early enough, she knows, 'I went to the clinic, I was told I cannot deliver normally'. She will come here on time. So, she will be told, 'The moment you have reached 40 weeks, go to the hospital', she will be here. We do

her C-section very safely; it is very simple she goes home. NBU decongested here … also the chorio-amnionitis are no longer there. (R012, Department In-Charge)

*HCWs were reducing the time they spent with each mother.* Given the heavy workload that the HCWs were facing, they were reducing the time they allocated to providing each mother with care. This reduction in time meant that some lower-level facilities had to send away mothers due to the higher numbers of patients they were receiving:

Owing to the fact that the patient numbers are higher than the health workers, the burden on the health worker is greater. Meaning the time allocated per patient is less than required. (R005, Facility-Level Manager)

### Element 3: management and organisation
#### Availability of essential physical resources
*The policy has improved the availability and standards of equipment and supplies.* With the help of reimbursements from the free policy, the facilities reported to have had improvements in the availability of supplies and medical equipment. In fact, the facilities have kept reordering supplies to keep up with the demand:

… we've not actually gone out of stock. But you find we have to keep reordering because the demand is more. (R020, Facility-Level Manager)

Further, it was shown that with the availability of equipment and supplies, the HCWs did not have to use substandard care or equipment. For instance, one facility showed how they had now departmentalised equipment sterilisation process, rather than using the hospital steriliser. With this came the availability of delivery packs, and they are no longer using ordinary blades as before:

… we have so many like delivery packs which we used not to have. Sometimes we used to …. Use a blade instead of a delivery pack or the scissors because these things were not there …. There are people who are employed to cater for washing those things … and take …them (to) utility for preparation for next use. (R014, Nursing Officer)

*The facilities had improved infrastructure due to LM.* Some facilities used the reimbursements from the policy to improve infrastructure (such as upgrading their theatre and ultrasound areas). Additionally, some were expanding their buildings to reduce congestion. For instance, one facility was able to complete a section of a previously incomplete building, and transferred mothers from the congested postnatal ward to the new section:

… when our mothers are many in this maternity (in facility C) … those without complications or those who had delivered yesterday, we transfer them to that department, so there is that decongestion. And we have another building there, the reproductive

health, it is only that it is not yet over (complete) … but now the patients who are being attended … were transferred to that department and …we got the extension. (R014, Nursing Officer)

Other facilities even renovated older buildings that were no longer in use and converted them into maternity clinics to ease congestion. For instance, in facility B, one building constructed 5 years ago to be a mortuary but was only used to store patients' records, was refurbished and transformed into an outpatient clinic. The downside of this transformation, was that the mothers had a negative attitude towards the clinic, as they still perceived it as a mortuary.

Additionally, the policy reimbursements helped facilities meet their essential services, which were critical in easing the burden of work. As noted by HCWs, these reimbursements allowed them to incentivise mothers by using elements such as transport that would help improve quality of care. However, with the increase in patients, there was a corresponding increase in workload:

… sometimes that money will help to fuel the vehicle and … to maintain the ambulance … (and) sometimes it can support … staffs to go for seminars and … to conduct those in-reach … and also outreach services. (R008, Nursing Officer)

… in a way it's a pusher to more quality service to the client … because you want … to attract more … because the more, the better. But … that also has brought the issue of us bursting through the seams. (R019, Facility-Level Manager)

*Enhanced facility resources and facility characteristics.* The women in the EI ascertained that there was an enhancement of the resources in the facilities due to the policy. The facilities were shown to have adequate waiting and examination rooms (51.60%); adequate hand washing facilities (91.82%); adequate bathing facilities (67.46%); adequate toilet facilities (71.45%); well-suited equipment for detecting women's problems (90.91%); had an adequate number of staff (76.37%) who are well suited to treat women (96.55%) and had an overall clean environment (93.45%) (online supplemental appendix 4). However, the mothers showed some concern about the adequacy of the facility providing clean drinking water, as indicated by 46.18% of mothers (online supplemental appendix 4).

*Despite progress, some infrastructure, commodities and supplies are still a challenge to some facilities.* Some respondents noted that some facilities still have inadequate medical equipment (such as ultrasound machines), space and supplies. The lack of these basic elements, such as a basic laboratory, was demotivating the women from using the services in the hospital and preventing HCWs from completely following up with the mothers as they would have wished to.

… we don't have a very vibrant laboratory … as a clinician, I believe you want the patient tested, drugs availed, that patient will not come back to you after two days (said with wry humour). You can give them a prescription, and they tell you they bought half a dose because they didn't have money, now, how will you help them? You see, it demotivates …. Yes. Even the ultrasound, the scans, we don't have the scans, so they have to do the scans outside (the facility) …. About the (ward) it's not an ideal labour ward. We don't even have an ideal resuscitaire, you know, the improvised one? … you have to be extra cautious not to shake that thing, so the heater falls on the baby. Imagine, you have three mothers delivering, and you deliver as you put there …. In the process you can burn those babies as you go to pick the other one … so, you have to be extra cautious …. Even IPC (infection prevention and control) becomes an issue. (R018, Facility-Level Manager)

The noted challenge regarding the supplies was that the county government was prioritising improving infrastructure, which was visible to the women, and seen as a better investment, rather than providing supplies and medication. The HCWs noted that providing medication posed the biggest challenge, possibly due to the drug ordering protocol. The facilities had to wait for a certain number of days before receiving top up for their orders:

… there is a protocol … because like our drugs are ordered through KEMSA for a certain period, by any chance those drugs are not enough … they get finished before that period, we have to wait for the other order. But usually, in a hospital like ours (high-level facility), sometimes we are given extra money like miscellaneous where you can purchase emergency. But even when you purchase emergency like drugs, we are able to purchase a start dose or a prophylaxis, for continuity, you find now you have to involve maybe the patient. (R020, Facility-Level Manager)

## Competent and motivated staff

*Mothers have a strong positive perception of healthcare delivery characteristics by the HCWs.* A majority of the mothers in the EI had a positive perception of the healthcare delivery characteristics. For instance, 95.27% perceived that the staff examined pregnant and postpartum women well; 95.45% noted that the staff were very capable of finding out what is wrong with mothers; 59.64% noted that staff prescribed drugs that are needed and that the drugs supplied by the health facility were good (58.37%) and the mothers could obtain the drugs from health facility easily (67.27%) (online supplemental appendix 4). In addition, 71.27% perceived that they received adequate information on danger signs of delivery and postpartum (online supplemental appendix 4). Interestingly, 79.82% perceived that the facility provided privacy during vaginal examination and delivery and 84.70% believed that the

procedure they received during ANC and delivery felt very much necessary (online supplemental appendix 4).

*Nurses multitask and handle many roles among human resource challenges.* Nurses, especially in the lower-level facility, were shown to go over and above in their work. They covered night and day shifts, handling other hospital consultations at night and still accompanied the pregnant women during referrals. Despite the tasks being their roles, the constrained number of nurses was making the staff rotation allocation challenging, and hence, they had to multitask among the challenges.

Besides, because of the challenges of the increased workload from the LM policy, even the nurses in charge of both department and hospital administrations found themselves pulled into hands-on nursing practice, leaving their office doing administrative work, to ensure timely service provision. Also, workload in the maternity wing reached an overwhelming point, requiring the nurses to seek help from other departments:

> … there is also the issue of shortage. Like today, we are so many, but at least we have covered all areas. But other times we report like three people, so … we have to work here and go to that place (to work in the wards). (R007, Department In-charge)
>
> We call help from other departments when it's so much. (R001, Department In-Charge)

*HCWs are adequately motivated to work despite the challenges they face.* The HCWs reported being motivated to work more because they believed putting more efforts into providing services, would result into more LM reimbursement funds to the facility, which would subsequently translate to better services and additional staff (through locum nurses):

> … the policy of Linda mama has motivated the staff. At least we know that if you put more effort, there will be more funds on the facility, we will get more commodities, we will be compensated for escort (referral) and lunch … it will be more comfortable for us. (R003, Nursing Officer)

The hospital in-charges noted that despite the high workload, they believed that the HCWs remained motivated and that they presented a perfect picture during supervision. For example, they noted that some were even comfortable running the wards alone without the support of other nurses, and some were even forfeiting their lunch time:

> … they go overboard (HCWs) … you would find two nurses on night duty, conducting 15–17 deliveries … alone. And finding this nurse has to monitor this mother from admission, delivery and postnatal and also the baby, you find they go overboard … like our nurses in maternity, they would not even break for lunch. They would wait until now the shift is over. (R020, Facility-Level Manager)

Some mothers reported that the HCWs attended to them, even outside their working shifts, which signified dedication to work:

> R4: I came here at 2:00 pm, and I got a doctor who was on the morning shift and the other one was changing. So, I told him to serve me, I wanted to deliver. He dressed in a hurry and came to help me. (Woman in FGD003)

In fact, the other cadre of HCWs, such as department clerical officers, noted that among the challenges, they worked beyond the stipulated hours either to support the provision of LM services or to batch the claims and ensure that the hospitals receive timely reimbursement. However, they faced a challenge with inadequate and insufficient infrastructure (such as computers to ease work) and salaries. However, despite these challenges, they perceived that they knew how to plan their days and work.

*HCWs' source of motivation was not solely driven just money.* Some of the factors that the HCWs indicated as a source of motivation, rather than monetary values, were the kind acts and listening ear of the county administration and facility in-charges. For instance, in one facility, the department in-charge felt that the administration heard and acted upon their grievances, including renovating the theatre and expanding the admission area. Others also felt that this resulted in the provision of adequate equipment and supplies to the facilities without having to improvise the outdated equipment:

> … at least we are listened to when we at least raise something … at least we get better service operating because of that. I mean theatre … was moved from here the squeezed area to that place, and then there wasn't bed, it was brought. (R001, Department In-Charge)
>
> … once in a while, we call them, have breakfast meetings with them, listen to their issues, discuss with them. (R016, County Senior-Level Manager)

The other source of motivation was that HCWs were happy when their burden of work was eased and department in-charges were doing so by employing additional people on locums, providing training opportunities and recognising them for risking their lives at night during referrals to other facilities.

Furthermore, the nurses felt that they were involved in decision making and they perceived that it gave them a voice to raise an opinion on how the work needs to be done:

> So that one I see at least they could have involved us the people on the ground. (R014, Nursing Officer)

*There were some causes of demotivation and dissatisfaction among HCWs.* For instance, HCWs noted that they felt inadequately remunerated despite the increased workload occasioned by the policy. With the workload, others

felt that they had to multitask (for instance, handle referrals at all hours of the night and still had to come back to the facility after referral to carry out their duties, and which they felt they were not adequately motivated for):

> We are underpaid, yeah let me say that without fear because we do a lot of work. You see like the time you came into the office; I was so buried there. I have been sitting there since 7:30 am. (R011, Clerical Officer)

Similarly, the in-charges of the maternity departments, who were also HCWs, noted that the delayed reimbursements from the LM policy demotivated them. With such delays, the in-charges were having a strained working relationship with the hospital suppliers and even banks:

> You are doing your services, and you are claiming, but you are not getting the benefit of your work, so it renders even demoralising the people (HCWs in) the maternity … the same might demoralise even the suppliers who do supply us with the goods … some of them do cut off deals with dealing with the facility. Because we do pay them very late, and sometimes, they attract interest in their banks. (R006, Nursing Officer)

### Monitoring and continuous quality improvement

*Nurses monitor the QoC provided through partographing and charting labour progress, though they face challenges.* The nurses showed awareness of proper documentation of labour progress using a partograph to enhance quality care. However, they noted that they sometimes faced additional challenges (presentations/conditions from the patients, for example, those from referrals or mothers who came in at the second phases of labour and delivered within a few minutes of admission) that they did not know how to document.

although once in a while a file maybe there is a problem, but they try … because you know a partograph is very important … I know maybe you have found challenges in those partographs when you were going through. (R007, Department In-Charge).

Despite the challenges, the nursing in-charges and facility managers were organising additional education for staff on the monitoring processes for pregnant women. The university students, who were posted to the facilities for training, or even nurses who had had more recent training, were tasked to provide additional education to the nurses as they had more up-to-date knowledge.

### Element 4: experience of care

Overall, a majority of the mothers (84.2%) from the EI were completely satisfied with the services they received (hospital A (85.1%), B (80.9%) and C (85.2%) were completely satisfied with the services provided). A higher proportion of mothers in hospital C (74.4%) than B (66.7%) and A (74.1%), would consider future delivery in the same health facility (figure 2).

### Effective communication with the patients

*Mothers in the study had a positive experienced and perception of the HCWs.* A majority of the mothers in the EI had a positive perception (agreeing and completely agreeing) about the HCWs as being very open (94.34%); compassionate (90.58%); respectful (95.46%); devoted adequate time to the mothers (94.18%) and are very honest (92.00%) (online supplemental appendix 4). Some mothers noted that the HCWs were empathetic, friendly and reassuring. They appreciated the additional good treatment and sacrifices the HCWs made, such as warming food and additional support (such as bathing the baby and changing bedsheets and stained beddings) following the exhausting birth experience.

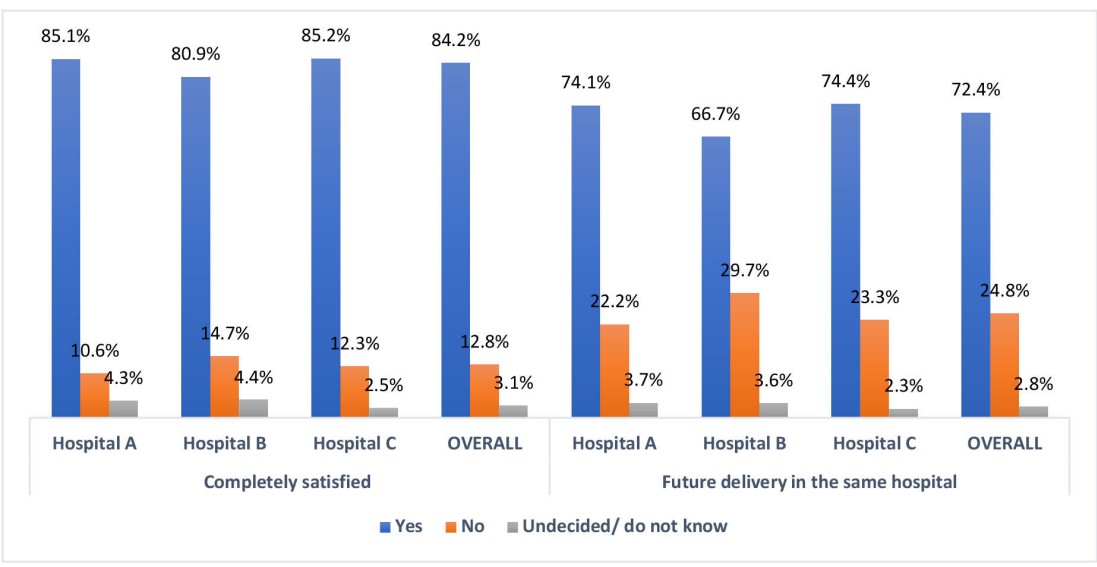

**Figure 2** Overall satisfaction and future delivery.

Some mothers appreciated being given priority in treatment, especially during emergencies by the doctors. In such circumstances, the firmness and decisiveness of the nurses were also perceived positively as being intent on preserving the lives of both the mother and baby. One mother was particularly impressed with the doctors who called for assistance in emergency scenarios when they were not able to handle them at the time:

R1: when I came once I got a certain doctor and I think there was an emergency, and I was forced to wait but I did not take offence because … he called another doctor who came here and I saw they have experience because they just serve you. (Woman in FGD003).

*Mothers were happy to have received information about emergency/procedures, and training on breastfeeding, family planning and baby care.* Some mothers highlighted that since some doctors explained to them the medical procedures they were to undergo, they were able to relieve some anxiety around birth especially:

R2: The doctor was good, he told me how it (procedure) would be done, and I was good. (Woman in FGD004)

The nurses supported the mothers during breastfeeding, taught them how to breastfeed and even encouraged those with difficulties. Some hospitals even went further by demonstrating to the mothers through *YouTube* videos, the breastfeeding procedures, which they (mothers) perceived as very useful and helpful. The facility in-charges acknowledged that they trained and empowered the nurses with breastfeeding knowledge to ensure that they in turn train the mothers:

And once this nurse trains in the breastfeeding, she'll go back, we make it as a duty for her to be educating the mother on those … on breastfeeding. (R020, Facility-Level Manager).

Besides breastfeeding, the mothers acknowledged being taught about family planning, how to wash the baby's cord and what to do if the baby faced some complications, which they considered reassuring.

*Inadequate preparation for birth by the HCWs.* Some HCWs were perceived as not being well prepared to handle the birth of the baby, given that they never had the birth equipment readily laid or that some materials and supplies were not readily available. This ultimately resulted in birth complications such as amniotic fluid aspiration.

*The lack of proper education and communication on expectations.* Some mothers felt there was no clear communication on the immediate care after delivery, which created a knowledge gap and resulted in potential mistakes with medications, that led to medical emergencies. For instance, one mother indicated:

R6: For my child there was a time I put the Hexi-cord (cord cleaning medication) on their nose. I did not know; I asked my husband to pass me the medicine at night thinking it was a nose drip. So, we thought that was it and we administered to him, we were forced to bring the baby here at night. (Woman in FGD001)

Some HCWs were perceived as not being reassuring and unable to provide mothers with the expected reassurance:

R12: 'We have examined you; the baby is not close'. You know, sometimes you feel the baby is close, and when it's time to deliver, many doctors and nurses came and told me, 'Why are you disturbing us, you are standing on the floor. Climb the bed'. I could not climb. They said, 'We are referring you to (a referral hospital)'. Now I said, oh my god what will I do? At that time, they started to insult me and told me, 'Come here, you are going to deliver in the ward'. (Woman in FGD009)

### Respect and dignity

*Food was perceived to be inadequate in some hospitals.* Some mothers revealed that despite having a good birth experience in the labour ward and did not pay for the delivery, the food provided by the support staff post-delivery, was inadequate, untimely, and unwholesome. Some mothers in some facilities highlighted that appetence for food could sometimes last for a whole night post-delivery, and, thus they resorted to having their relatives and family bring them food.

R2: I didn't pay anything, though their food is too little for a pregnant woman. It's true, it's too little, a mother has delivered, that food … and then they serve it very early, when it reaches 9pm you are hungry again…Yes, I had to call home (for food) because I felt weak. R7: There was a day I stayed here without food the whole night. I wasn't given. (Women in FGD009)

However, the administration revealed that the instance of food inadequacy may have been caused by the support staff who, despite the facility planning for adequate food for all hospital patients, may have rationed the food further. Despite the inadequacy of food, some mothers acknowledged that the food was actually good:

R3: Yeah, it was good, I ate good things, and even the bathroom was clean. The services there are good. (Woman in FGD003)

*There was overcrowding and bed-sharing, leading to a lack of privacy (congestion) and essential equipment and supplies, decreasing the QoC.* Congestion in the maternity department, as a result of the expanded FMP was a significant issue, especially in the higher-level facilities. The lower-level facilities equally faced an increase in the number of mothers, particularly for ANC and delivery, although the mothers did not share beds:

R3: but the problem I found here is congestion …. The first three hours (following CS) … I slept on a bed alone, after three hours we were two people on the bed. And from there the room we were taken too we would sleep four people with children, six people like that in one bed …. Because I left there with a back problem because I cannot sleep, you are forced to sit, you sit for the child to sleep. (Woman in FGD005)

Nonetheless, the hospitals gave bed priority to mothers who had underwent a caesarean section (CS) over those who gave birth normally. These mothers were allowed to sleep alone on the bed (space-permitting), in addition to having a special monitoring room. In contrast, mothers who had given birth normally were forced to share beds with other mothers, or made to sleep on the floor, with their babies being the only ones given beds. The congestion in the public facilities, therefore, forced the mothers to seek care elsewhere.

Also, despite the expanded FMP, the lack of basic essential equipment and space contributed significantly to poor QoC, even in maternal and child health clinics for PNC:

Go to MCH (maternal and child health clinic) … and see how babies are weighed naked outside, in this harsh weather. It is at times very cold in the morning but what do we do, we have to weigh them … but we are glad that we are still able to offer services. (R018, Facility-Level Manager)

### Emotional support

*Women experienced physical, verbal and emotional abuse.* Some mothers experienced both physical and verbal abuse from HCWs and support staff. The abuse was exacerbated by the lack of clarity in communication with HCWs. For instance, one woman reported that the nurses had slapped her for being stubborn and uncooperative during birth. Another woman mentioned that the nurse had tried to suture her episiotomy without using anaesthesia, and still another one received abuse in return from either support staff or HCWs for requesting support:

R6: I was slapped here …. For being stubborn; R3: you see someone is still in pain, they do not inject you with anaesthesia and they want to stitch you. Things like that are not good, this is also a human being, and they still feel pain. R5: I saw someone who had gone through a CS, and they told the nurse, they wanted to rise up, you know there is pain while rising up … but I saw her telling that nurse to help her get up, I saw (heard) the nurse insult her and I did not like that. (Women in FGD001)

Equally important was one mother's testimony showing how she was wheeled to the theatre in an undignified and uncaring manner:

R5: What I saw, what he did to me, when I was experiencing labour pains, I was told to go to theatre, and I told him I cannot walk. He pushed me like a cart up to the theatre. I told him I could not walk; he pushed me like a lorry. (Woman in FGD008)

*Some mothers experienced a lack of attention/care, negligence and unhygienic practices from the HCWs and support staff.* For instance, in one case, a doctor was shown to have forgotten to remove cotton wool used in packing blood after delivery:

R6: Like in my case they did not remove that thing (cotton wool) and then I went home with it. (Woman in FGD002)

Additionally, some mothers perceived that some HCWs were not giving them and their babies proper attention while attending to them and they felt ignored. For instance, one respondent whose baby required medical oxygen felt a lack of support:

R6: the baby came out fine. But I saw that by the time the nurse received him, he wasn't breathing well and then the nurses did not care because when I woke up after six hours I had to go look after my baby, when the oxygen came out, I would put it back, I changed everything. So, this time round I did not like them. (Woman in FGD006)

Some mothers were subjected to unhygienic practices by some HCWs, including being examined on an unclean bed previously used by another patient without cleaning it, or being left unattended for long:

R4: Another thing that I didn't like there, you are examined on a bed that someone else had been examined on and it is damp. It wasn't good. Like for me I was examined on a bed that had some liquid substance; R9: I delivered at (a referral hospital); I didn't like their services at all. Because when I delivered, I was cut down there (episiotomy) and the doctor left me for 30 minutes. On coming back, he stitched me with all that dirt, so I was not happy at all with their service. (Women in FGD009)

Some support staff also exacerbated the unhygienic practices of the mothers. For instance, one mother noted:

R6: when I delivered here, I was asleep, when I woke up around 6.30. I found they (support staff) had opened windows as they wanted to clean. If you had put your bag on the floor, they ask you to pick it up and put it in bed and that bed is where you place the baby, and the ground is dirty. (Woman in FGD001)

### DISCUSSION

To the best of our knowledge, this is the first study to examine the Quality of Care (QoC) across the continuum of maternal care (antenatal, perinatal and postnatal care) under the expanded Free Maternity Policy (FMP) or the Linda Mama (LM) policy in Kenya.

Our findings show that the LM policy reduced geographical access barriers, by harnessing more private sector and faith-based facilities to enhance service provision. Additionally, it eliminated financial access barriers through the incentives of free maternal care and increased utilisation of maternal services (more mothers seek skilled birth attendance; reducing home deliveries). These findings are consistent with the results from systematic reviews of maternal services under different free maternity policies, which showed increased maternal (ANC and delivery) services after removing user fees.[60 61] Dossou et al[62] also showed a systematic increase in caesarian section (CS) services after implementing the CS policy in Benin due to utilisation incentives. However, the reviews showed that the utilisation patterns under free policies were marred by geographical and temporal fluctuations in use, which differs from our study.

Furthermore, despite the policy enhancing access, the facilities were using additional approaches and incentives to attract mothers, leading to a difference in perception of the services provided. The finding on factors leading to the choice of the delivery place is not new, as other authors have highlighted the difference in the preference for private or public facilities thus influencing perception.[63–65] In fact, in a recent FGD with women in Nairobi's informal settlements in Kenya, exploring their experiences of the quality of maternity care under LM, Oluoch-Aridi et al[66] present the facilitators and barriers to choosing health facilities, which are all similar to the findings of this study. Interestingly, the choice of delivery facility was influenced by several factors that are not necessarily related to LM, such as personal choice, previous experience or treatment and access, as shown in other studies,[4 67] or health system factors.[68] This highlights a key gap because it raises the question of whether LM has influenced the choice of hospital for delivery. Escamilla et al[69] showed that the need for free services in Kenya had influenced women to bypass nearer facilities for farther private facilities that offered free care, which is similar to the findings from Sierra Leone by Fleming et al.[70]

Our finding shows that although there was an increase in the utilisation of free maternal services, the health facilities and HCWs had to bear the burden of providing service to more mothers seeking labour and delivery care. This finding is consistent with other studies, which have shown a significant increase in the utilisation of maternal services following the implementation of the free policy in Kenya,[14 71 72] which was attributed to the removal of cost barriers for women.[73] However, our study goes further to highlight that despite bearing the burden, the facilities and the HCWs were shown to be working beyond their capacity to provide care, leading to burnout among healthcare workers. This unintended consequence of the increased burden on the HCWs could be explained by the fact that the implemented policy did not translate into an equal investment in an increased number of HCWs, thereby exacerbating the burden. Previous studies have shown that the perennial lack of human resources

has always been a problem in Kenya. For instance, Miseda et al[74] reveal that out of the 138 266 HCWs required to fit the MoH Norms and Standards Guidelines for service delivery, only 31 412 are employed at the public sector, private facilities, and faith-based organisations (FBOs).

It was also shown that the HCWs went beyond their strengths to serve the increased number of mothers, with the aim of maximising reimbursements from the LM policy. However, this could cause burnout if not followed by an increased workforce, thus leading to poor QoC. Two meta-analysis studies have shown that HCWs burnout could lead to the provision of poor QoC.[75 76] HCWs are motivated by what Franco et al[77] deriving from Herzberg et al[78] refer to as 'hygiene factors' (determining HCWs dissatisfaction) in this case the interpersonal relationship with the county and the administration, and 'motivating factor' (determining HCWs motivation and satisfaction) in this case being heard. However, the facilities struggled to employ specialists, and there were other HCWs staffing challenges.

Our study has highlighted the enhanced identification strategies for vulnerable populations (such as street children, orphans and adolescents) that had initially been excluded from the policy on paper and are now using the policy. The findings align with the results of implementing the Safe Motherhood programme in Nigeria (Abiye initiative), which equally showed that removing user fees, particularly for the most vulnerable population, enhanced access and utility of service.[79] However, in a different study in Kenya, researchers showed that the enhancement of the reach of the vulnerable population was mainly done by HCWs who, bound by ethics and professionalism, provided expanded FMP services to those excluded from the policy, such as foreigners, and those without IDs, such as street children who had no parents, refugees without IDs or schoolgirls who were underage and pregnant. Hence, there is a need for official policy correction.[80] While our results further show that there has been enhanced equity and financial access to the services by the women as those in the rural and urban areas received uniform services for free, in Benin, the CS policy exacerbated the inequalities as the policy reached the predominantly rich, exacerbating social exclusion.[62]

Besides, from our findings, there is a positive perception of the policy despite the longer waiting times, particularly in the initial visits where mothers are accessing ANC additional benefit packages that were not in the previous policy. In contrast, a mixed-method study in Nigeria showed that mothers were dissatisfied with the waiting time under the free policy, but the authors did not link it to any particular service.[81]

A rather interesting finding is the mothers' preference for higher-level facilities due to the perception of better services. Higher-level facilities are significantly burdened due to LM policy, leading to a ripple effect (where the facilities are left with a resource gap, as they use more resources to meet the mothers' specialised needs and manage deliveries that can be done at the periphery).

 

However, it could also be argued that having more mothers in higher-level facilities means more claims and reimbursements. However, literature has attributed this preference to factors such as cleanliness, interpersonal skills and other perceptions of better services[82] and not the LM policy. A discrete choice experiment in Nigeria showed that the women chose to give birth in places with good condition of the health system, and absence of sexual, physical and verbal abuse and that an unclean environment of birth without privacy and unclear user fees policy drove the women away.[83] The mother's choice of higher-level facilities has led to QoC concerns such as indifference in the treatment based on the type of delivery and parity (partly because of overburdening higher facilities and the need for prioritisation). In Kenya, other studies have shown that mothers bypass lower-level facilities due to the perception of better quality.[84 85] Same case has been shown in Sri Lanka.[86]

Interestingly, fewer mothers are being referred from lower to higher facilities than before the LM policy. While in the previous policy, complications were being referred to higher-level hospitals from lower-level health centres to seek better services,[87] it could be argued that, through the LM policy, lower-level facilities are making adequate investments using the LM policy reimbursements and are thus able to handle complications. That may nevertheless not be true as another finding in our study showed that the fewer referrals are mainly due to the lack of equipment, theatre and NBU in the lower-level facilities. Thus, it could be that the policy confusion in the reimbursements of the services is somewhat hampering the positive quality effects of the policy. Other literature from Ghana concurs with this assumption. For instance, Witter *et al*'s[88] exploration of the policy showed that the uncoordinated and unreimbursed referral strategy (particularly at referring hospitals) hampered the positive effect of the policy, while Ganle *et al*[89] showed that Ghana's referral system was ineffective and the care was substandard because of a lack of critical care staff to handle healthcare emergencies.

The mothers who are referred have a positive perception of the referral process. This perception could be because the HCWs went above and beyond to provide referral elements, such as allowing the mothers to have companions at referral time and in the hospital. However, the lack of transport for referral could hamper the referral gains by either making the mother pay or risk their life looking for transport systems at the tail end of delivery. For example, Burkina Faso included transport in their subsidy policy to enhance mothers' referrals to health facilities.[90] Through its well-organised rapid response to emergency and evacuation, mothers were positively satisfied with the referral system under the policy; however, IDIs with HCWs revealed no adequate follow-up to ensure the evacuated mothers received care as intended.[91] Interestingly, Kenyan nurses under the LM policy went above and beyond to refer and follow-up mothers, which was a compensatory mechanism for improving QoC.

In addition, through the LM policy, there has been some improved availability of equipment, supply and infrastructure. The improvement could be due to the provider and facility in-charges using Streel-Level Beureacrat tacts (such as renovations) to improve the facility and hence to attract more mothers who are the source of reimbursement funds.[80] However, despite progress, some commodities, infrastructure and supplies remain a challenge. The lack of supplies, equipment and infrastructure contravenes the WHO statement number eight on quality, which shows that positive birth outcomes rely on their availability.[41] A recent review showed that inadequacy is a global phenomenon compromising the quality of maternal care.[92] Evidently, in all the facilities, the mothers revealed that they were satisfied with the characteristics of the facilities, such as having adequate rooms, adequate hand washing, bathing and toilet facilities; in addition to equipment well suited for detecting women's problems. As is in this study, a mixed-methods study in Ghana showed that, despite the inadequate infrastructure in the facilities and lack of basic supplies, 89% of the mothers who participated in the EI, and those in the FGDs were satisfied with the quality of maternal care during childbirth[93] as is in this study. This postulates that mothers are more concerned about the interpersonal care received and the basic amenities provided if they can have live births and remain alive. The absence of or inadequacy of equipment and supplies compromises the QoC.

Equally interesting was that the good experience of care received by the women was based on the level of support provided by the HCWs and the facilities. Research shows that a good relationship between patients and HCWs could help improve trust, diffuse patients' anxieties, and create open communication.[94] The majority of the mothers in both the FGDs and the EI attributed the good experience of care to the interpersonal skills exhibited by the HCWs, such as empathy, being friendly, kindness, respect, devoting time and honesty. The good care experiences the women receive influences their future delivery in the same facility. However, the findings could not show whether such experiences were due to LM policy, except that it incentivised the HCWs to provide FP and breastfeeding education. The finding shows that despite the challenges of the policy, the mothers appreciated and perceived the HCWs and health facility characteristics positively. This shows that HCWs have significantly contributed to the provision of care, but this may not lead to improved outcomes if the technical aspects of quality are not met. Similar findings have been reported elsewhere where, for instance, in Ghana, 77% of the mothers who participated in the EI noted that they were content with the HCWs service provision as they were patient and empathetic[93] or in Ethiopia, where 79.1% of the mothers interviewed were happy with the overall services provided.[95]

The poor experience of care by the mothers hampers the technical QoC received. By sharing the beds due to overcrowding, the mothers are exposed to unhygienic

practices that could eventually lead to nosocomial infection in the maternity facilities, which hampers QoC. A review of quality elements in facilities in the 14 counties in Kenya linked the introduction of LM services with poor hygiene and low privacy[29] Such findings are expected because investments in hospital infrastructure have not subsequently followed the increase in the number of mothers utilising maternal care. Other literature has shown similar findings in other settings with FM services.[96–98]

The other finding of poor QoC experienced by the mothers, such as lack of attention, negligence and physical abuse, has been shown in other Kenyan literature. For instance, the beneficiaries of FM services in a study in Kakamega provincial hospital in Kenya noted that the HCWs negligence and use of vulgar language were demeaning to the patients.[99] Food is an important component in the birth process and for mothers to report that the food they received during delivery is inadequate is as surprising as it is demeaning. Also, as is in this study, poor communication with the mothers, or the lack thereof, may create an ethical dilemma, especially in contexts where patients do not consent to or have not been adequately informed about procedures.[100] Mothers should therefore be involved, to play a role in the decisions of the care they receive.

A key limitation of this study is that the EIs were conducted in one county, and it is plausible that there could be varied practices across other counties. The implication of this study is that it may be difficult to generalise the findings to all the other 47 counties in Kenya. Nonetheless, using IDIs and FGDs in this study provides an opportunity to unpack the issue at hand (quality of maternal care under LM policy) within its context and be analytically generalisable. The meta-issues identified by the study are likely to be found in other counties, even though they might manifest in different ways.

## CONCLUSION

This study has demonstrated that the Linda Mama policy has had a positive impact on the quality of care across all the broad quality domains: access to care (equitable and timely), provision of care (safe and effective), management and organisation and the experience of care. There were positive elements such as minimised access barriers (cultural, financial and geographic), timeliness of care and provider availability, which have created functional referral systems and safety, and availability of essential physical resources and competent and motivated staff. The women in the study had a good care experience, which included receiving prompt maternal services, good care for the baby after birth, teaching about birth procedures, breastfeeding and family planning. However, the study also revealed negative results due to the policy, such as the lack of supplies, equipment and infrastructure, and referral challenges, hampering maternal care. There were cross-cutting poor experiences by the women, such as overcrowding of the healthcare facilities, inadequate food supply, a lack of communication regarding treatment plans, and experiencing physical and verbal abuse. In order to achieve the SDG and UHC goals that seek to ensure reduced maternal morbidities and mortalities through access to quality service for every woman, it is crucial to address the negative aspects of the policy highlighted in this study, while strengthening the positives.

**Acknowledgements** The authors acknowledge the data collection support of the EI data from research assistants: Rachel Murigu, Justus Miran, Billy Bortich, Valentine Olunga, Janet Moraa, Winnie Kaitany and Shillar Jeptoo.

**Contributors** BO: conceptualised the study, curated and analysed the data, and drafted the initial manuscript which was subsequently revised for important intellectual content by all authors. ZA-P: contributed to drafting the initial manuscript, reviewed and edited all the drafts. SK and SP: contributed to the design and supervised the study. All authors read and approved the final manuscript. BO is responsible for the overall content as the guarantor.

**Funding** We gratefully acknowledge the financial support provided by the Commonwealth Scholarship Commission (KECS-2017-266), which supported BO's PhD study, on which the data and the results of this study are based, and the Economic and Social Research Council (grant number ES/P000622/1) for covering the publication costs of this study. The funding agencies did not play any role in the analysis, interpretation of the results or manuscript writing.

**Competing interests** None declared.

**Patient and public involvement** Patients and/or the public were not involved in the design, or conduct, or reporting or dissemination plans of this research.

**Patient consent for publication** Consent obtained directly from patient(s).

**Ethics approval** This study involves human participants. Ethical approval was obtained from the University of Kent, SSPSSR Students Ethics Committee and AMREF Scientific and Ethics Review Unit in Kenya (Ref: AMREF-ESRC P537/2018). Further written permission to conduct the study was received from the county government of Kiambu, and all the participating hospitals. Written and oral informed consent was obtained from the potential participants before starting the interviews. Participants gave informed consent to participate in the study before taking part.

**Provenance and peer review** Not commissioned; externally peer reviewed.

**Data availability statement** Data are available upon reasonable request. An anonymised, de-identified version of the dataset can be made available on request to allow all results to be reproduced. All requests should be directed to the corresponding author.

**ORCID iDs**
Boniface Oyugi http://orcid.org/0000-0002-9550-9138
Zilper Audi-Poquillon http://orcid.org/0009-0002-6901-7348
Sally Kendall http://orcid.org/0000-0002-2507-0350
Stephen Peckham http://orcid.org/0000-0002-7002-2614

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
