## [Reviewer comments · BMJ Open]

ARTICLE DETAILS

TITLE (PROVISIONAL)	Examining the quality of care across the continuum of maternal care (antenatal, perinatal, and postnatal care) under the expanded free maternity policy (Linda Mama Policy) in Kenya: a mixed-methods study
AUTHORS	Oyugi, Boniface; Audi-Poquillon, Zilper; Kendall, Sally; Peckham, Stephen

VERSION 1 – REVIEW

REVIEWER	Ochieng, Beverly Tropical Institute of Community Health and Development
REVIEW RETURNED	07-Dec-2023

GENERAL COMMENTS	It would be better if the authors provided a description of the quality of care across the continuum of maternal care (antenatal, perinatal, and postnatal) in the setting where the study was conducted (Kiambu County). The architecture of the free maternity policy in Kenya, in general, will lead to an understanding of the linkage being made with the specific services that are studied within this work. It's not clear from the abstract what analysis techniques were used to collect and analyse the data for this research study. The study site should be clearly stated in the abstract, and this should be boldly highlighted in the background section too. It would be better if the authors provided a description of the problem (backed by statistical evidence) in the setting where the study was conducted (Kiambu County). Based on the choice of the study site, this should be a basis to expound on the problem statement. What studies have been conducted in Kiambu County highlighting the issues? These will help in giving a clear description of the study title too. It's also important to analyse broadly, in the introduction, the UHC-defined benefit package in the free maternity package, how the service is integrated into the global benefits package, and its impact on the quality of care across the continuum of maternal care. Kenya has a total of 47 counties with diverse demographics and social and cultural characteristics. Looking at the study site, the research was conducted in Kiambu County. It's not clear why only one county was selected. Is the Kiambu County results a true picture when generalising the study results to Kenya as a whole?
---

	It is important to see the extent to which free maternity policy is affecting the quality of care, and that should be discussed in the introduction for any comparison that could be made later in the discussion. It's not clear how saturation was achieved in these interviews or how we can say that the opinions being expressed fully represent reality (link with the selection criteria that should be done within a strategy to cover all aspects that are presented in the objectives list but also avoid biases) Have you shared the answers with your interviewees before sending them for publication? If not, how can you be sure that you have transcribed the right opinions?
--	--

REVIEWER	Ombere , Stephen Okumu Maseno University, Sociology and Anthropology
REVIEW RETURNED	15-Dec-2023

GENERAL COMMENTS	Thank you for this interesting timely manuscript. However, I have some concerns which if incorporated will strengthen this article before publication. I consider them suggestions and some comments seeking clarification to make sentences clear. Though in the table and list of abbreviation, It is important to kindly write the data collection methods in full before abbreviating them see Page 7, under study population, sampling, and data collection. How many IDIs and when did the authors stop conducting IDIs? Below are my detailed comments. On the Title: Since this article talks about Linda Mama I strongly suggest the authors add "expanded" just before free maternity for the title to read.....under the expanded free maternity policy in Kenya. on the abstract: I am not sure whether you want to include an expanded free maternity policy (EFMP) If yes then affect the changes in the entire article. I encourage the authors to provide the current statistics of Kenya's MMR on line 19 on page 4. On page 4 on line 37 authors need to review additional literature and cite other authors who have studied the implementation of free maternity in Kenya. This will strengthen this article. On page 4 Linda mama is abruptly introduced. On line39-42 the idea does not flow. Authors should build a strong foundation for LM and link it with Universal Health Coverage (UHC) before proceeding to the next paragraph. Notably, there is no link between the paragraph ending on line 42 and the new paragraph on line 44. The authors can revise this for coherence by incorporating my suggestions herein. There is a good attempt to show what exists however, much needs to be done. For instance, Line 24 page 3, line 7 page 4 authors need to diversify the literature search to have a synthesis of what other scholars have researched. However, there is a mixup between Linda Mama and UHC. There is no background upon which the UHC agenda is built in this
--

	article. Additionally, the authors need to have a seamless flow of ideas addressing the two issues and show a point of intersection. The framework of Analysis is adequate and fits very well for this study. The choice for Kiambu County is not scientifically convincing. The authors can revise it by expunging redundant issues such as 'logistics feasibility of data collection, proximity to Nairobi County' etc. Retain/include issues that make Kiambu County unique from Mandera, Wajir, Kisumu counties etc that made it unique and outstanding for this study. In ethics consideration...replace the word "bigger study" with larger study. Page 9 line 39-41, the authors to state exactly how these are ensured in this article rather than reporting what informed the participants about. What do authors mean by "error! Reference source not found" in the findings section? Kindly rectify this. Sign-post statement in the discussion section is misleading. There are authors who have conducted studies under free maternity though not so much on the expanded free maternity services. Revise this section. Discussion section adequately articulates the emerging themes from this study. However, some sections require strengthening and clarity to show what new the study is contributing. For instance, line 52-60 on the discussion section. I would be happy to read this article again. Wish you all the best.
--	--

VERSION 1 – AUTHOR RESPONSE

Reviewer: 1

Dr. Beverly Ochieng, Tropical Institute of Community Health and Development

Comments to the Author:

It would be better if the authors provided a description of the quality of care across the continuum of maternal care (antenatal, perinatal, and postnatal) in the setting where the study was conducted (Kiambu County).

We thank the reviewer for the comment. In the last paragraph under the introduction, we have introduced the the continuum of maternal care (antenatal, perinatal, and postnatal) concept based on previous literature. Still, we have gone further under the study setting section of the methods and discussed the Kiambu county setting in detail in paragraph two. Elements of maternal care have also been included in the framework of analysis under the methods section.

The architecture of the free maternity policy in Kenya, in general, will lead to an understanding of the linkage being made with the specific services that are studied within this work.

We thank the reviewer for this comment. As suggested by the reviewer, we have built a strong foundation for the architecture of the free maternity policy in Kenya and its linkage to specific services in paragraphs 2, 3 and 4 in the introduction section.

It's not clear from the abstract what analysis techniques were used to collect and analyse the data for this research study.

We thank the reviewer for this comment. We have now added the analysis techniques in the abstract. It now reads, 'Quantitative data was analysed descriptively while qualitative data was analysed thematically. All the data were triangulated at the analysis and discussion stage using a framework approach guided by the QoC for Maternal and Newborn.'

The study site should be clearly stated in the abstract, and this should be boldly highlighted in the background section too. It would be better if the authors provided a description of the problem (backed by statistical evidence) in the setting where the study was conducted (Kiambu County). Based on the choice of the study site, this should be a basis to expound on the problem statement. What studies have been conducted in Kiambu County highlighting the issues? These will help in giving a clear description of the study title too.

We thank the reviewer for this comment. This study was conducted across multiple levels within the Kenyan health system. We have stated so in the abstract. It now reads, "We conducted a convergent parallel mixed-methods study across multiple levels of the Kenyan health system, involving key informant interviews (KIIs) with national stakeholders (n=15), in-depth interviews (IDIs) with County officials and healthcare workers (HCWs) (n=21), exit interview survey with mothers (n=553) who utilised the LMP delivery services, and focus group discussions (FGDs) (n=9) with mothers who returned for postnatal visits (6, 10, and 14 weeks)."

We have expounded the description of the study setting in the methods section. In it, we have described the problem (backed by statistical evidence) in the study's setting (Kiambu County). We have now expounded, under the section study setting in the methods, the reason for the choice of Kiambu County, including its sociodemographic characteristics, health indicators, and population size.

It's also important to analyse broadly, in the introduction, the UHC-defined benefit package in the free maternity package, how the service is integrated into the global benefits package and its impact on the quality of care across the continuum of maternal care.

We thank the reviewer for this comment. In the second, third and fourth paragraphs, we have revised the section and created a background upon which the UHC agenda is built in this article. Further, we have provided the UHC-defined benefit package in the free maternity package and how the service is integrated into the global benefits package. No study has ever shown the impact of the benefits package on the quality of care across the continuum of maternal care, and we believe that this is not linked to this study but is another important subject upon which future studies can be built. In this line,

we have only shown the one study on the quality of maternal care across the continuum in the last paragraph of the introduction and linked it to this study.

Kenya has a total of 47 counties with diverse demographics and social and cultural characteristics. Looking at the study site, the research was conducted in Kiambu County. It's not clear why only one county was selected. Is the Kiambu County results a true picture when generalising the study results to Kenya as a whole?

We thank the reviewer for this comment. We have now expounded, under the section study setting in the methods, the reason for the choice of Kiambu County, including its sociodemographic characteristics, health indicators, and population size. We have gone further and linked the reason for the linkage of this study to a larger referenced study but still incorporated the logistic feasibility of data collection as of the time of data collection. We have explained this further.

Yes, Kenya indeed has a total of 47 counties with diverse demographics and social and cultural characteristics. We have highlighted this limitation under the limitations at the end of the discussion section. However, while the results may not be generalisable beyond the study county (area) because of the heterogeneity of the counties, we have identified significant contextual factors that may have influenced the patterns of implementation and the findings which are transferable (enhance transferability) to other 47 counties in the counties and can be used to interpret the implications of the results in other settings. We believe that the use of mixed methods permitted for complementarity and triangulation of the information across multiple levels within the Kenyan health system, and these results are imperative for the implementation experiences across the counties and the country.

It is important to see the extent to which free maternity policy is affecting the quality of care, and that should be discussed in the introduction for any comparison that could be made later in the discussion.

We thank the reviewer for this comment. As suggested by the reviewer, we have now discussed the extent to which free maternity policy affects the quality of care in paragraphs 3 and 4 in the introduction section and under the study setting in the methods section.

It's not clear how saturation was achieved in these interviews or how we can say that the opinions being expressed fully represent reality (link with the selection criteria that should be done within a strategy to cover all aspects that are presented in the objectives list but also avoid biases)

We thank the reviewer for this comment. We have now added how saturation was achieved under the Study population, sampling, and data collection section in paragraph 3. We have shown that the KIIs and IDIs were stopped when there was no new information forthcoming or further dimensions, nuances, or insights of issues found.

Have you shared the answers with your interviewees before sending them for publication? If not, how can you be sure that you have transcribed the right opinions?

We thank the reviewer for this comment. Following the ethics review recommendation and the requirements from the study sites, all the transcripts of the KIIs and the IDIs were shared with the respondents to ascertain the contents before analysis. Further, the researchers shared the study findings with the county and the study sites (in line with the agreements) for feedback and validation. The FGDs were never shared with the respondents, but during analysis, the researchers, not only

listened to the recording for familiarisation, but also read and reread the transcripts to make sure they captured the actual recordings.

Reviewer: 2

Dr. Stephen Okumu Ombere , Maseno University

Comments to the Author:

Dear Authors,

Thank you for this interesting timely manuscript. However, I have some concerns which if incorporated will strengthen this article before publication. I consider them suggestions and some comments seeking clarification to make sentences clear.

Though in the table and list of abbreviation, It is important to kindly write the data collection methods in full before abbreviating them see Page 7, under study population, sampling, and data collection. How many IDIs and when did the authors stop conducting IDIs?

We thank the reviewer for this observation. At the onset of the section Study population, sampling, and data collection we have now written the data collection methods (Eis, FGDs and IDIs) in full before abbreviating them and at the tail end of Table 1, we have added a notes section on the abbreviations used in the table (Eis, FGDs and IDIs) that shows what they mean for ease of the reader. Other abbreviations throughout the document that were not fully indicated in the first instance have now been corrected.

We have now added the numbers of KIIs and IDIs that were conducted. Further, we have shown that the KIIs and IDIs were stopped at the point where no new information was forthcoming or no further dimensions, nuances, or insights of issues were found.

Below are my detailed comments.

On the Title:

Since this article talks about Linda Mama I strongly suggest the authors add "expanded" just before free maternity for the title to read.....under the expanded free maternity policy in Kenya.

We thank the reviewer for the suggestion. We have now amended the title as suggested.

on the abstract: I am not sure whether you want to include an expanded free maternity policy (EFMP) If yes then affect the changes in the entire article.

We have made this change as suggested by the reviewer so that there is a good way to distinguish the two FMPs (2013 and 2017).

I encourage the authors to provide the current statistics of Kenya's MMR on line 19 on page 4.

We thank the reviewer for this comment. However, We believe we have captured the latest statistics on Kenya's MMR, which can also be found here:

<https://www.who.int/publications/i/item/9789240068759>. We would be happy to update this figure if the reviewers can share newer statistics.

On page 4 on line 37 authors need to review additional literature and cite other authors who have studied the implementation of free maternity in Kenya. This will strengthen this article.

We thank the reviewer for this comment. Having distinguished between the 2013 and 2017 policies in the introduction, we have now captured additional literature as suggested by the reviewers. We agree that this gives a new perspective and enhances the work flow.

On page 4 Linda mama is abruptly introduced. On line39-42 the idea does not flow. Authors should build a strong foundation for LM and link it with Universal Health Coverage (UHC) before proceeding to the next paragraph. Notably, there is no link between the paragraph ending on line 42 and the new paragraph on line 44. The authors can revise this for coherence by incorporating my suggestions herein.

We thank the reviewer for this comment. As suggested by the reviewer, we have now built a strong foundation for LM and its linkage with UHC before proceeding to the next paragraph. Following the reviewer's suggestion, we have linked the two subsequent paragraphs discussing the literature on FMP.

There is a good attempt to show what exists however, much needs to be done. For instance, Line 24 page 3, line 7 page 4 authors need to diversify the literature search to have a synthesis of what other scholars have researched.

We thank the reviewer for this comment. In the introduction section, we have added and diversified the literature to strengthen the arguments using what other scholars have researched. While we are cognisant that this has significantly added to the article's word count, we believe it makes the flow better for ease of understanding.

However, there is a mixup between Linda Mama and UHC. There is no background upon which the UHC agenda is built in this article. Additionally, the authors need to have a seamless flow of ideas addressing the two issues and show a point of intersection.

We thank the reviewer for this comment. In the second, third and fourth paragraphs, we have revised the section and created a background upon which the UHC agenda is built in this article. We believe that this has created a flow of the two ideas. UHC agenda has been published extensively in the context, and we believe we have limited our element of UHC only to fit the scope of this article.

The framework of analysis is adequate and fits very well for this study.

We thank the reviewer for this positive comment.

The choice for Kiambu County is not scientifically convincing. The authors can revise it by expunging redundant issues such as 'logistics feasibility of data collection, proximity to Nairobi County' etc. Retain/include issues that make Kiambu County unique from Mandera, Wajir, Kisumu counties etc that made it unique and outstanding for this study.

We thank the reviewer for this comment. We have now expounded, under the section study setting in the methods, the reason for the choice of Kiambu County, including its sociodemographic characteristics, health indicators, and population size. We have gone further and linked the reason for the linkage of this study to a larger referenced study, which still incorporates the logistic feasibility of data collection as of the time of data collection. We have explained this further.

In ethics consideration...replace the word "bigger study" with larger study.

We thank the reviewer for this comment. Under the ethics consideration section, we have now actioned the change as suggested.

Page 9 line 39-41, the authors to state exactly how these are ensured in this article rather than reporting what informed the participants about.

We thank the reviewer for this comment. In table 2, under the functional referral system section, we have now actioned the change as suggested.

What do authors mean by "error! Reference source not found" in the findings section? Kindly rectify this.

Apologies, this error message came from dissociating the tables from the main text. These instances have been replaced with the correct table and appendices numbers in brackets throughout the document.

Sign-post statement in the discussion section is misleading. There are authors who have conducted studies under free maternity though not so much on the expanded free maternity services. Revise this section.

We thank the reviewer for this comment. We have made this adjustment as the reviewer suggested under the discussion section's signpost. It now reads 'under the expanded FMP or LM policy in Kenya.'

Discussion section adequately articulates the emerging themes from this study. However, some sections require strengthening and clarity to show what new the study is contributing.

For instance, line 52-60 on the discussion section.

We thank the reviewer for the positive comment in the first part of this point. To enhance clarity in the discussion section that the reviewer has mentioned, we have rewritten the whole paragraph afresh so that there is more clarity to what our study is contributing to.

The section now reads: 'Interestingly, our finding shows that while there was an increase in the utilisation of free maternal services, the facilities and HCWs bore the burden of providing service to more mothers seeking LM care. This finding aligns with other authors' findings, which have shown that there was a significant increase in the utilisation of maternal services following the implementation of the free policy in Kenya, which was attributable to the removal of cost barriers to women. Nonetheless, our study goes further to highlight that despite bearing the burden, the facilities and the HCWs were shown to be working beyond their capacity to provide care to the extent that the HCWs ended up experiencing burnout. The unintended consequence of the increased burden on the HCWs could be explained by the fact that the implemented policy did not translate to an equal investment in an increased number of HCWs, hence the burden.'

I would be happy to read this article again. Wish you all the best.

We thank the reviewer for this positive affirmation.

VERSION 2 – REVIEW

REVIEWER	Ochieng, Beverly Tropical Institute of Community Health and Development
REVIEW RETURNED	06-Mar-2024

GENERAL COMMENTS	The manuscript has been revised and corrections have been have been incorporated.
---

REVIEWER	Ombere , Stephen Okumu Maseno University, Sociology and Anthropology
REVIEW RETURNED	15-Mar-2024

GENERAL COMMENTS	Thank you Editors for the second chance to review this article. I am so impressed with the effort authors' have put in to revise this manuscript. Despite good effort, I would like to ask authors to read through this manuscript to weed out typos and spelling mistakes which are notably minor. I only have one major concern that authors need to attend to "I raised a concern about the justification for the choice of study area. The authors have endeavoured to address it however, could the authors delete information on lines 6-10. they can simply say, Kiambu county was purposively selected for this study considering the fact that the county has been shown to pose higher trends in maternal mortality.....compared to other counties around Nairobi from central region..." Lastly, under study population, kindly consider deleting "Delete meaning on line 50". After clarifying issues raised above, this article is will be adequate in form and content. I would be happy reading this article finally published. I wish the authors all the best.
---

VERSION 2 – AUTHOR RESPONSE

Reviewer: 1

Dr. Beverly Ochieng, Tropical Institute of Community Health and Development

Comments to the Author:

The manuscript has been revised and corrections have been have been incorporated.

We thank the reviewer for this positive comment.

Reviewer: 2

Dr. Stephen Okumu Ombere , Maseno University

Comments to the Author:

Thank you Editors for the second chance to review this article. I am so impressed with the effort authors' have put in to revise this manuscript.

We thank the reviewer for this positive comment.

Despite good effort, I would like to ask authors to read through this manuscript to weed out typos and spelling mistakes which are notably minor.

We thank the reviewer for this comment. We would like to confirm that we have now re-read the manuscript and weeded out the typos and spelling mistakes.

I only have one major concern that authors need to attend to

"I raised a concern about the justification for the choice of study area. The authors have endeavoured to address it however, could the authors delete information on lines 6-10. they can simply say, Kiambu county was purposively selected for this study considering the fact that the county has been shown to pose higher trends in maternal mortality.....compared to other counties around Nairobi from central region..."

We thank the reviewer for this comment. We have now corrected the line as suggested by the reviewer. It now reads "Further, Kiambu County was purposively selected for this study because the county has been shown to pose higher trends in maternal mortality compared to other counties around Nairobi from the Central Region."

Lastly, under study population, kindly consider deleting "Delete meaning on line 50".

We thank the reviewer for this comment. We have elected to clarify this statement so that we do not lose the meaning of 'meaning saturation'. It now read 'The KIIIs and IDIs were stopped at the point where no new information, further dimensions, nuances, or insights were forthcoming (i.e. when meaning saturation was attained)'.

After clarifying issues raised above, this article is will be adequate in form and content. I would be happy reading this article finally published.

We thank the reviewer for this positive comment.

I wish the authors all the best.